# Assessment of Force Production in Parkinson’s Disease Subtypes

**DOI:** 10.3390/ijerph181910044

**Published:** 2021-09-24

**Authors:** Paulo Henrique Silva Pelicioni, Marcelo Pinto Pereira, Juliana Lahr, Paulo Cezar Rocha dos Santos, Lilian Teresa Bucken Gobbi

**Affiliations:** 1School of Physiotherapy, Division of Health Sciences, University of Otago, Dunedin 9016, New Zealand; 2Posture and Locomotion Studies Laboratory, Universidade Estadual Paulista (UNESP), Rio Claro 13506-900, Brazil; mppereir@yahoo.com.br (M.P.P.); ju_lahr@hotmail.com (J.L.); paulocezarr@hotmail.com (P.C.R.d.S.); ltbgobbi@rc.unesp.br (L.T.B.G.); 3Department of Computer Science and Applied Mathematics, Weizmann Institute of Science, Rehovot 7610001, Israel

**Keywords:** Parkinson’s disease, subtypes, muscle power, force production

## Abstract

Muscle weakness is a secondary motor symptom of Parkinson’s disease (PD), especially in the subtype characterized by postural instability and gait difficulty (PIGD). Since the PIGD subtype also presents worse bradykinesia, we hypothesized that it also shows a decreased rate of force development, which is linked to an increased risk of falling in PD. Therefore, we investigated the effects of PD and PD subtypes on a force production profile and correlated the force production outcomes with clinical symptoms for each PD subtype. We assessed three groups of participants: 14 healthy older adults (OA), 10 people with PD composing the PIGD group, and 14 people with PD composing the tremor-dominant group. Three knee extension maximum voluntary isometric contractions were performed in a leg extension machine equipped with a load cell to assess the force production. The outcome measures were: peak force and rate of force development (RFD) at 50 ms (RFD50), 100 ms (RFD100), and 200 ms (RFD200). We observed lower peak force, RFD50, RFD100, and RFD200 in people with PD, regardless of subtypes, compared with the OA group (*p* < 0.05 for all comparisons). Together, our results indicated that PD affects the capacity to produce maximal and rapid force. Therefore, future interventions should consider rehabilitation programs for people with PD based on muscle power and fast-force production, and consequently reduce the likelihood of people with PD falling from balance-related events, such as from an unsuccessful attempt to avoid a tripping hazard or a poor and slower stepping response.

## 1. Introduction

Parkinson’s disease (PD) is characterized by the presence of non-motor and motor symptoms, and reduced muscle function (strength and neuromuscular control) is one of the motor symptoms presented by people with PD [1]. Physiopathological aspects of PD may reflect motor control impairments and can therefore lead to deficits in force modulation and production [2]. Although the underlying mechanism of the deficiencies in force modulation and force control is not agreed upon, central aspects of these deficits are likely related to impairments in muscle function in PD [3]. For instance, impairments in basal ganglia structures reflect abnormalities in force recruitment [4], for example, by reducing the antagonist muscles’ activation followed by an augmented activation of antagonists muscles. Unsurprisingly, people with PD usually report muscle weakness, during which the peak of force is up to 50% lower than their healthy peers, depending on disease severity [5]. Muscle weakness emerges as a health concern since it is associated with reductions in functionality, such as poor mobility [6,7,8], deficient postural control [6], lower physical activity levels [9], and is also associated with an increased mortality rate [10]. Especially for PD, disease progression accompanies muscle weakness and may reflect some motor symptoms [3,5].

In contrast with pure maximal voluntary force, the rate of force development (RFD, defined as the derivative of the force production over time curves [11]) is better related to the performance of daily activities, and it is more sensitive to intervention and presence of chronic diseases [12,13,14]. Functionally, RFD is particularly relevant because it reflects the capacity to use muscles rapidly in response to different demands, including daily situations (e.g., avoiding an obstacle while walking [15] or recovering from an unexpected loss of balance [16]). Furthermore, evidence suggests that increased excitability in the motor cortex is associated with fast voluntary contraction [17]. Additionally, PD’s effects on the brain during a ballistic movement lead to impairments on the neural drive to muscle, partially explaining bradykinesia [18] and the diminished velocity of muscle contraction [2,14]. Therefore, it is reasonable that force development is associated with motor impairments in PD [2,19], mainly considering the previous literature linking reduced RFD and risk of falling in this population [20].

Specifically, bradykinesia is associated with reduced lower-limb muscle power in PD [21]. Interestingly, in previous studies, bradykinesia and quadriceps weakness were found as prevalent PD motor symptoms in a subset of individuals with idiopathic PD characterized by their marked postural instability and gait difficulty (PIGD) subtype [22,23], in comparison to individuals with PD categorized as tremor-dominant (TD) due to their marked tremor symptoms. In addition, albeit previous studies show worse bradykinesia, muscle weakness, gait instability, and a more significant number of falls in the PIGD subtype [22,23,24,25,26], muscle-power generation has not yet been investigated within different PD subtypes. Once this information is available, allied health professionals can plan therapeutic interventions focusing on this neuromuscular issue and muscle power generation to improve other motor functions in PD. Therefore, we aimed to investigate the effects of PD and PD subtypes on force production characteristics (RFD and peak force). We hypothesize that people with PD vs. older adults (OA) would exhibit impaired force production. In addition, the PIGD subtype would have impaired force production compared with the TD subtype.

## 2. Materials and Methods

We chose a convenience sample of thirty-eight participants for this study. We included fourteen OA participants in this study, recruited from an exercise program (*Programa de Atividade Física para a Terceira Idade*) at the São Paulo State University in Rio Claro, Brazil. We also included twenty-four participants with idiopathic PD who were recruited from “*Programa de Atividade Física para Pacientes com Doença de Parkinson*” at the Posture and Locomotion Studies Laboratory, Rio Claro, Brazil. Participants were eligible if they had a diagnosis of idiopathic PD, according to the UK PD Brain Bank criteria [27]. We assigned PD participants to two groups according to their clinical characteristics: TD (14 participants) and PIGD (10 participants)—described ahead. We considered OA participants if they did not have any history of neurological disorders nor any additional health issue that would preclude their ability to participate in the study. PD participants were eligible to participate if they had a diagnosis of idiopathic PD and if they did not present any additional health issues or neurological disorders that could preclude their ability to participate. Standard inclusion criteria included community-dwelling participants who could walk for 30 m without walking aids and stand up with no support for 1 min.

We collected participants’ information, such as sex, age, height, weight, and duration of the disease since the diagnosis, and Unified Parkinson’s Disease Rating Scale (UPDRS) part III (motor score) [28]. We used UPDRS for motor subtype classification according to the ratio of the mean tremor scores (part II, item 16 (tremor) and part III, items 20 (rest tremor scores) and 21 (action tremor scores)). We then used the mean of PIGD scores (part II, items 13 (falling), 14 (freezing), and 15 (walking), and part III, items 29 (gait) and 30 (postural stability)) in the calculation. Mean TD/mean PIGD was used to define subtypes: ratio ≥1.5 classified TD patients; ratio ≤1.0 classified PIGD patients; ratio between 1.01 and 1.49 classified indeterminate subtype (excluded from analyses; n = 2) [29]. We also collected information about antiparkinsonian medication intake as levodopa equivalence dosage (LED) daily intake [30]. We scheduled all assessments according to the optimal response of the medication in the “on” state of medication (approximately 1 h after medication intake).

We used a leg extension machine to examine force production and knee extension maximum voluntary isometric contraction. We attached a load cell (EMG System^®^, EMGSystem do Brasil, São José dos Campos, Brazil) perpendicularly to the movement axis to measure force production. Participants were seated in a backward inclined chair during the tests, with knee joint at 110° and hip joint at 90°, with feet resting in a plate (0° = full extension). After a single maximum voluntary isometric contraction for familiarization purposes, participants performed three maximum contractions of 5 s duration and 1 min rest between each attempt. We instructed the participants to “push as fast as you can, and then keep pushing using as much power as you can, until I tell you to stop”. Thus, the participants extended their knees as fast and hard as possible and maintained the maximal force for 5 s [31,32]. During the maximum contraction, the participants received verbal encouragement during each attempt [31]. Such verbal encouragement was standardized and equal among groups. Data were collected using a biological data conditioner (EMG System^®^) and specific software (WinDaq^®^, Dataq Instruments Incorp., Akron, OH, USA). We calculated the average of three attempts for each participant.

The outcomes measures were: peak force, identified as the maximum peak achieved during the maximum voluntary isometric contraction; we calculated participants’ rate of force development as the average slope of the force-time curve over 50 ms (RFD50), 100 ms (RFD100), and 200 ms (RFD200) after the force production onset [31]. We used specific Matlab^®^ coding (version R2012b, The Mathworcs Inc., Natick, MA, USA) for all data analyses.

We used SPPS for Windows (SPSS v.24, Inc., Chicago, IL, USA) for statistical analysis. We used univariate analysis of variance (ANOVA) or chi-square tests for cross-tabulation tests (binary categorical variables) to contrast demographic measures between all groups. We compared clinical measures between the TD and the PIGD groups using Student’s t-tests for independent samples (normally distributed variables), Mann−Whitney U-tests (non-parametric variables), or chi-square tests for cross-tabulation tests. We used ANOVAs to contrast peak force and RFD between groups (PIDG vs. TD vs. OA). We conducted unadjusted (LSD) post hoc analyses when we found significance in ANOVA. We used Cohen’s d to report the effect sizes, considering values equal to 0.2 (small), 0.5 (medium), 0.8 (large), and 1.2 (very large) [33]. We set significance levels at 0.05 for all analyses.

## 3. Results

Table 1 shows the demographic and clinical comparisons among groups. We did not find any statistical differences between groups (OA, TD, and PIGD) for demographic measures (*p* > 0.05). Likewise, we did not observe any statistical difference between the TD and the PIGD group for clinical measures (*p* > 0.05).

Figure 1 and Table 2 show the comparison between groups in force production. We observed statistical difference between groups in all variables (peak force, F_2,36_ = 7.04, *p* = 0.003; RFD50, F_2,36_ = 14.35, *p* < 0.001; RDF100, F_2,36_ = 12.72, *p* < 0.001 and RDF200, F_2,36_ = 9.83, *p* < 0.001). Post hoc analysis indicated that, compared with the OA, the TD and PIGD groups exhibited lower peak force (*p* = 0.032 and <0.001, d = 0.82 and 1.66, for TD and PIGD, respectively), RFD50 (*p* < 0.001 for both PD sub-groups, d = 1.48 and 2.19, for TD and PIGD, respectively), RDF100 (*p* < 0.001 for both PD sub-groups, d = 1.37 and 2.15, for TD and PIGD), and RDF200 (*p* = 0.003 and <0.001, d = 1.11 and 2.04, for TD and PIGD, respectively). No statistical difference was observed between PD subtypes (TD and PIGD) for RFD50, RFD100, and RFD200 (*p* > 0.05). The statistical significance remained the same for all comparisons even after controlling for covariates, such as age, sex, body mass index (BMI), and Hoehn and Yahr (HY) stages (for PD only) (Table 2). Figure 2 shows a simple representation of force versus time for each group and each RFDs.

## 4. Discussion

We confirmed our hypothesis that people with PD would exhibit impaired force production when compared with OA. Lower peak force and RDF50, RDF100, and RDF200 were observed in people with PD, regardless of the subtype, compared to OA. Even after controlling for covariates, such as age, sex, BMI, and HY stages (for PD only), these results remained the same. Regarding our hypothesis, the PIGD subtype would exhibit impaired force production compared with the TD subtype, we observed that PIGD vs. TD subtypes showed lower maximum force values and developments (Peak Force and RDF50, RDF100, and RDF200). However, these differences were not statistically significant.

As previously reported in the literature, the force produced by muscles depends on the motor unit (MU) recruitment and rate coding (rates at which motor neurons discharge action potentials) [32]. Impairments on neuromuscular control, such as those evidenced in people with PD, affect the capacity to recruit motor units [34] and, consequently, reflect lower force production and rate developments. Therefore, it is not surprising that a comparison between PD (combined both TD and PIDG) OA indicated an approximate 28% lower peak force and over two-fold slower RFD (Figure 1). Our results quantitatively corroborate previous literature that showed lower RFD in people with PD compared with healthy controls [2,14,34]. The incapacity of people with PD in recruiting many single MUs simultaneously with a high firing rate within a relatively limited time [14,34], mainly of agonist muscle [35,36], may underlie the evident muscle weakness and reduced capacity to produce rapid/ballistic movement. This incapacity might be attributed to basal ganglia impairments in PD since this neural structure is essential for inhibiting unneeded and inappropriate muscle activation [37]. Extrapolating our results to functional tasks, muscle weakness and reduced force development are associated with poor functional performance in the general population [38,39,40] and people with PD [21]. These deficits are more evidenced in people with PD because impairments accompany muscle weakness and deficiencies in the RFD in functionality, impaired muscle strength, and its development, and might, at least, partially explain the evident deficits in the mobility of these individuals, such as on obstacle avoidance and sit-to-walk [15,16,41]. These deficits also explain the difficulty in sustaining repetitive muscle contraction as found in previous research [42], which may increase the risk of falls in people with PD compared to their healthy peers [16].

Although we observed a trend indicating lower peak force and RFDs in PIGD vs. TD (Figure 1), we did not observe statistical differences between PD subtypes. The PIGD subtype exhibits more bradykinesia [22] and risk of falls [23], and both bradykinesia and falls are associated with reduced lower limb muscle power [21]. While we did not observe significant differences between subtypes, a subtype comparison indicated a slight reduction in peak force and force development (small to medium Cohen’s d effect size—0.40–0.58) for the PIGD subtype. Therefore, these trended differences may somewhat support the idea that impaired neuromuscular control and specific disease symptoms, such as bradykinesia, are accompanied by impaired muscle function in PIDG subtypes. Further studies are required to investigate such speculation and include motor unit analysis (e.g., electromyography).

Regardless of relevant results, we need to be cautious in interpreting these results since no movement was involved in the leg extension test, where movement amplitude and speed were not present. Both are essential to quantify bradykinesia. Thus, we should further explore the RFD in addition to tasks that require changes in movement amplitude and movement speed to confirm these results. In addition, we should carefully interpret and not purely attribute differences in specific motor symptoms in PIGD to impaired force production since we observed a trend with small to medium effect size. Still, statistical differences were not evident, albeit PIGD indicated greater ranges of differences in force production than TD when confronted with OA. Limitations also include: (1) we only assessed force production on quadriceps muscles. We understand that verifying the force production on other muscles such as hip abductors, knee flexors, and ankle dorsiflexors is essential because they are related to different functional tasks and risk of falling [20]; (2) we only measured participants in “on” medication status. Force production may differ among PD groups once levodopa is prescribed to improve bradykinesia and tremors [27,43]; (3) we undertook a static analysis that does not fully apply to daily life movements; (4) the participants enrolled in this study were recruited from a larger project involving different studies; therefore, the sample size of subgroups was unequal and relatively small (TD n = 14 and PIGD n = 10 participants). Thus, future studies should: (i) analyze different muscle groups; (ii) assess participants in both statuses “off” and “on” of antiparkinsonian medication; (iii) assess individuals undertaking a leg extension test, where movement amplitude and movement speed are present since both are essential to quantify bradykinesia; (iv) use electromyography to understand force production in these populations better; (v) consider larger sample sizes, sample sizes calculation, and adjustments for multiple comparisons accordingly.

## 5. Conclusions

This is the first study to investigate force production in different PD subtypes. In addition, people with PD exhibited lower RFD than OA, which might be due to bradykinesia and might be associated with basal ganglia deficits, which contribute to lower agonist activation, which impairs force production in a different extension. Together, our results indicated that PD affects the capacity to produce maximal and rapid force. Based on the results of this study, we should consider interventions based on muscle power and fast force production in rehabilitation programs for people with PD. These programs might lead people with PD to produce a faster response and consequently reduce the likelihood of people with PD falling from balance-related events, such as from an unsuccessful attempt to avoid a tripping hazard or a poor and slower stepping response.

## Figures and Tables

**Figure 1 ijerph-18-10044-f001:**
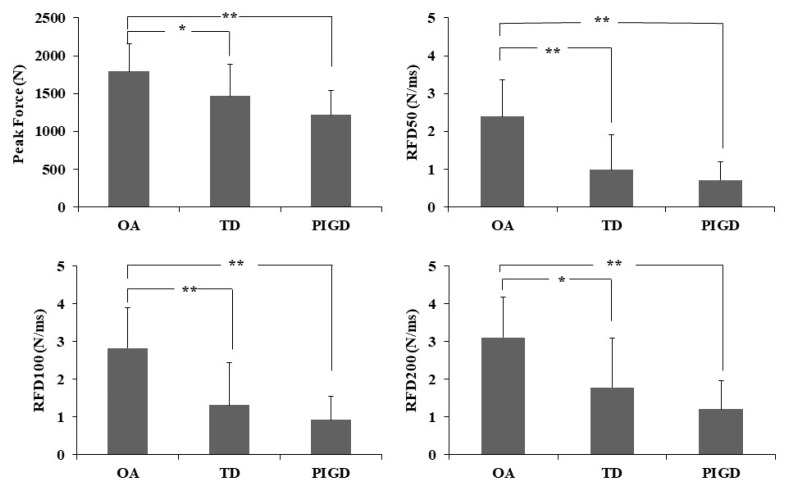
Force production comparisons between the OA, TD, and PIGD groups. Data are mean (SD). OA: older adults; TD: tremor-dominant group; PIGD: postural instability and gait difficulty group; RFD: rate of force development.* *p* < 0.05 and ** *p* < 0.001.

**Figure 2 ijerph-18-10044-f002:**
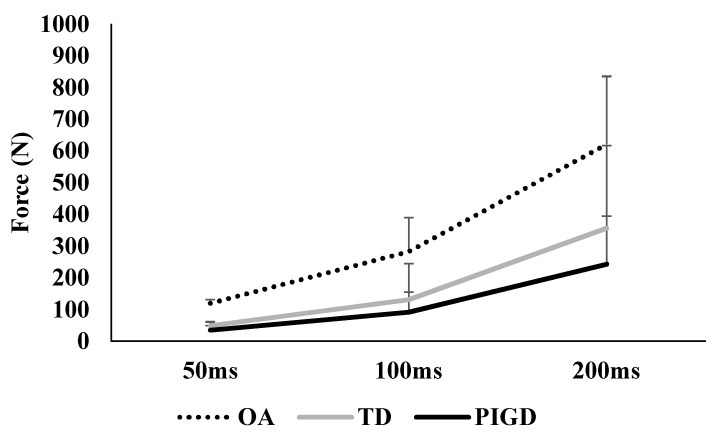
Representation of force versus time for each group and different rates of force developments (50 ms, 100 ms, and 200 ms). Data are mean (SD). OA: older adults; TD: tremor-dominant group; PIGD: postural instability and gait difficulty group.

**Table 1 ijerph-18-10044-t001:** Demographic and clinical measures per group. Data are mean (SD) unless stated otherwise.

Outcomes	OA (n = 14)	TD (n = 14)	PIGD (n = 10)	*p*
Sex Male (%)	9 (60)	11 (79)	6 (60)	0.498
Age (years)	69.7 (5.0)	72.0 (5.5)	70.1 (4.2)	0.436
Weight (kg)	67.7 (15.1)	75.1 (9.4)	64.2 (11.3)	0.092
Height (cm)	162.4 (8.0)	164.3 (5.9)	159.2 (8.0)	0.257
Disease duration (years)	-	5.1 (2.7)	6.4 (4.6)	0.452
UPDRS part III (score)	-	28.5 (9.0)	29.1 (7.3)	0.769
HY (stage)	-	1.9 (0.3)	2.0 (0.3)	0.596
LED (mg)	-	692 (478)	867 (509)	0.403

OA: older adults; TD: tremor-dominant group; PIGD: postural instability and gait difficulty group; UPDRS: Unified Parkinson’s Disease Rating Scale; HY: Hoehn and Yahr stage; LED: Levodopa equivalence dosage.

**Table 2 ijerph-18-10044-t002:** Force production comparisons between the OA, TD, and PIGD groups. Data are mean (SD).

Outcomes	OA (n = 14)	TD (n = 14)	PIGD (n = 10)	*p*
Peak force (N)	1791 (367)	1467 (417)	1217 (324)	0.003 ^a,b,c^
RFD50 (N/ms)	2.39 (0.97)	0.98 (0.93)	0.71 (0.49)	<0.001 ^a,b,c^
RFD100 (N/ms)	2.82 (1.08)	1.31 (1.13)	0.92 (0.63)	<0.001 ^a,b,c^
RFD200 (N/ms)	3.10 (1.07)	1.78 (1.30)	1.21 (0.75)	<0.001 ^a,b,c^

OA: older adults; TD: tremor-dominant group; PIGD: postural instability and gait difficulty group; RFD: rate of force development. ^a^ Statistical difference between OA and TD (*p* < 0.05); ^b^ statistical difference between OA and TD (*p* < 0.05); ^c^ all statistical significances remained the same after controlling for age, sex, BMI, and HY stages (for PD only).

## Data Availability

Other researchers can obtain access to the data after consultation with the corresponding author.

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
