# Peer review of "Assessment of Force Production in Parkinson’s Disease Subtypes"

_ijerph, 2021, doi:10.3390/ijerph181910044_

Round 1

Reviewer 1 Report

The article under review is entitled “Force production is affected in people with Parkinson’s disease”

The authors designed a study to examine rate of force development across PD “subtypes” and compared to healthy older adults. Three unequal groups were assessed via a “leg press”. The authors found that RFD and force production was lower in those with PD compared to controls. However, the authors did not detect any difference between the 2 PD groups.

The general theme that those with PD have diminished force capacity and RFD has saturated the literature. The novel aspect of the study suggested there were no differences in force and RFD between the “symptom dominant” groups. Unfortunately, the authors did not adequately address this in the discussion and how it could be added to treatment plans or fills a clinical significantly gap in the literature.

There were several methodological issues, that could influence the results and not information to justify the results.

The title did not adequately represent the manuscript and is very ambiguous.

Specific comments are below:

Comments:

Abstract: It is unclear what your results mean to the caregivers, rehab specialist, or neurologist.

Introduction:

line 27: the authors state; “(PD) is mainly characterized by the presence of motor symptoms”. This statement can be misleading when non-motor symptoms, i.e., speech and swallow difficulty, cognitive decline, and autonomic function, can present before motor symptoms. Change the phrasing to appropriately reflect this.

Line 29:  the authors should be careful when introducing the idea of motor control being responsible for reductions in force production. The authors need to properly justify this in order to suggest this.

Methods

Line 99: what was the rationale for choosing isometric measurement. If it is the authors hope to compare this to functional outcomes, which appear to be more dynamic nature, what is the rationale for this.

Why was knee extension the only muscle measured?

What was the rationale for looking at these levels. Rate of force development at 50, 100, and 200ms.

Was this a bilateral test, unilateral, more affected, or less affected. If this was bilateral, did you use something like EMG to determine if the legs were contributing equally?

Why was impulse not examined. During dynamic functional “activities”, the change in momentum of a body/limb is directly proportional to impulse, and therefore impulse is arguably the most functionally relevant RFD measure that can be taken during the rising force–time curve of explosive isometric contractions.

Line 100: You use the term leg press to describe the instrumented device however what you describe in a few lines lower is a leg extension machine. Please correct this. As Leg Press is a multi-joint machine.

Line 105. The authors utilize a 5 sec window to collect force data. However, the authors mention earlier that those with PD have trouble recruiting MU at a sufficient rate and so MVIC may not be reflected in this time window. Additionally, how the participants were asked to generate force is unclear. Asking to generate as fast possible, or just to push as hard as possible could influence the results significantly.

Based on Figure 1 you did not normalize the force output to individual’s body mass. Although there was no difference between body mass (p=.09) it is close to be different and based on SD it is possible for force data to be affected. Absolute force values do not accurately reflect individual characteristics this investigation focuses on. What was the authors rationale for this?

Line 109: The authors mention “Such verbal encouragement was standardized and equal among groups. However, it is unclear if the researchers blinded to the participants subtype. This would introduce bias to the results.

Line 110: The authors mention “Data were collected using a biological data conditioner (EMG System®) I assume this system had a load cell. If the equipment had EMG capability, why was it not presented. Gathering information on possible discharge rate, activation, how RFD correlated with muscle activity would be important addition to the study.

In table 1 of the results I did see any information on clinical characteristics: of the subtypes. You mentioned Mean TD/ mean PIGD ratios. Why were they not included?

Discussion

Line 157: The authors state “The force produced by muscles depends on the motor unit (MU) recruitment and rate coding (rates at which motor neurons discharge action potentials). The authors did not measure this and so it needs to be clear that this is speculative.

Line 176: “These deficits also might explain the difficulty in sustaining repetitive muscle contraction.” This is not warranted in your document since it was not measured

Author Response

The article under review is entitled “Force production is affected in people with Parkinson’s disease”. The authors designed a study to examine the rate of force development across PD “subtypes” and compared it to healthy older adults. Three unequal groups were assessed via a “leg press”. The authors found that RFD and force production was lower in those with PD compared to controls. However, the authors did not detect any difference between the 2 PD groups. The general theme that those with PD have diminished force capacity and RFD has saturated the literature. The novel aspect of the study suggested there were no differences in force and RFD between the “symptom dominant” groups. Unfortunately, the authors did not adequately address this in the discussion and how it could be added to treatment plans or fills a clinical significantly gap in the literature. There were several methodological issues that could influence the results and not information to justify the results. The title did not adequately represent the manuscript and is very ambiguous.

Thanks for your comments. We revised the manuscript, including the discussion. We also modified the title to avoid ambiguity.

Specific comments are below:

Comments:

Abstract:

It is unclear what your results mean to the caregivers, rehab specialist, or neurologist.

We revised the abstract to make it clearer for caregivers, rehab specialists and neurologists.

Introduction:

line 27: the authors state, “(PD) is mainly characterized by the presence of motor symptoms”. This statement can be misleading when non-motor symptoms, i.e., speech and swallowing difficulty, cognitive decline, and autonomic function, can present before motor symptoms. Change the phrasing to appropriately reflect this.

Our idea was to be brief and directly introduce the issues people with PD living with motor symptoms presented. However, we modified the sentence above not to neglect the existence of non-motor symptoms. The sentence now reads, “Parkinson’s disease (PD) is characterized by the presence of non-motor and motor symptoms, and reduced muscle function (strength and neuromuscular control) is one of the motor symptoms presented by people with PD”.

Line 29:  the authors should be careful when introducing the idea of motor control being responsible for reductions in force production. The authors need to properly justify this in order to suggest this.

We modified the sentence using words such as “may” not to make assumptions. Although we speculate that is the case, using the literature as backup and also because impairments in motor control can lead to deficits in force modulation and force production. This sentence now reads, “Physiopathology aspects of PD may reflect motor control impairments, therefore can lead to deficits in force modulation and force production”.

Methods

Line 99: what was the rationale for choosing isometric measurement. If it is the authors hope to compare this to functional outcomes, which appear to be more dynamic in nature, what is the rationale for this.

There are two rationales for the choice of isometric measurement. First, we acknowledge that using a Biodex, for example, could represent a better measurement of force production. We could also have used force plates; however, the time is very limited to the propulsion force exerted, and consequently, we would not be able to measure the rate of force development. The second reason to choose this test is that this could be reproduced in clinics and gyms if clinicians have a dynamometer or a cell force. This measurement is easier, more feasible and more affordable and could be easily assessed in clinical practice. We addressed this in the limitations section.

Why was knee extension the only muscle measured?

Because of equipment limitations, this was a small study. Also, our volunteers receive several invitations to participate in different studies; therefore, we keep our questions focused on one topic not to burden and overwhelm them. It would be interesting to measure other muscle groups associated with fall risks, such as ankle dorsiflexion and hip flexors. We also addressed these suggestions in the paragraph we discuss possible limitations.

What was the rationale for looking at these levels. Rate of force development at 50, 100, and 200ms.

The rationale is that we need to produce a faster response, for example, to perform such a volitional response, such as avoid a trip. Since such response seems to be impaired in individuals with PD, and for such reason, they are more likely to fall; we decided to use these rates of force development. Also, these rates were used in previous research by one of the authors. Please, see Pereira and Gonçalves, 2010, in the reference list.

Was this a bilateral test, unilateral, more affected, or less affected. If this was bilateral, did you use something like EMG to determine if the legs were contributing equally?

We measured force production bilaterally. Unfortunately, we did not use EMG. We believe the use of EMG would contribute to our results. We also addressed that as suggestions for future studies.

Why was impulse not examined. During dynamic functional “activities”, the change in momentum of a body/limb is directly proportional to impulse, and therefore impulse is arguably the most functionally relevant RFD measure that can be taken during the rising force–time curve of explosive isometric contractions.

We decided not to use impulse because our movement was isometric, without change in the range of movement. It would be interesting indeed, for example, to investigate impulse in tasks that require a change in range of motion, such as lifting a foot in response to a sudden obstacle on a participant’s walkway. We appreciate the comment from the reviewer, though.

Line 100: You use the term leg press to describe the instrumented device; however, what you describe in a few lines lower is a leg extension machine. Please correct this. As Leg Press is a multi-joint machine.

Thanks for clarifying this term. We modified the use of “leg press” throughout the manuscript.

Line 105. The authors utilize a 5 sec window to collect force data. However, the authors mention earlier that those with PD have trouble recruiting MU at a sufficient rate and so MVIC may not be reflected in this time window. Additionally, how the participants were asked to generate force is unclear. Asking to generate as fast possible, or just to push as hard as possible could influence the results significantly.

We asked the participants to “push as fast as you can, and then keep pushing using as much power as you can, until I tell you to stop”. We modified this in the methods section. We hope this clarifies. Now it reads, “We instructed the participants to “push as fast as you can, and then keep pushing using as much power as you can, until I tell you to stop”. Thus, the participants extend their knees as fast and hard as possible and maintain the maximal force for 5s”.

Based on Figure 1 you did not normalize the force output to individual’s body mass. Although there was no difference between body mass (p=.09) it is close to be different and based on SD it is possible for force data to be affected. Absolute force values do not accurately reflect individual characteristics this investigation focuses on. What was the authors rationale for this?

We re-analyzed the data, including body mass as a covariate and the results did not change. Furthermore, as suggested by reviewer #2, we also re-analysed the data, including age, sex and BMI as covariates, and the results did not change. Therefore, we believe our results describe the real difference between people with PD and healthy controls, and no difference among subtypes, after controlling for covariates.

Line 109: The authors mention “Such verbal encouragement was standardized and equal among groups. However, it is unclear if the researchers blinded to the participants subtype. This would introduce bias to the results.

This study is from a secondary analysis of the studies Pelicioni et al., 2019 and 2020. Thus, we were not primarily thinking about this hypothesis during the data collection. In addition, we only analysed the UPDRS after our data was collected; thus, we believe that the subtypes did not bias us.

Line 110: The authors mention “Data were collected using a biological data conditioner (EMG System®). I assume this system had a load cell. If the equipment had EMG capability, why was it not presented. Gathering information on possible discharge rate, activation, how RFD correlated with muscle activity would be important addition to the study.

Regardless we used the biological data conditioner; unfortunately, we did not collect any EMG data. This is a limitation that should be addressed in future studies.

In table 1 of the results I did see any information on clinical characteristics: of the subtypes. You mentioned Mean TD/ mean PIGD ratios. Why were they not included?

We only provided data that was pertinent for the current study. In addition, we usually do not report the mean TD/ mean PIGD ratios, given that the subtypes were divided after calculating such ratios. Therefore, we do not want to reproduce redundant data.

Discussion

Line 157: The authors state “The force produced by muscles depends on the motor unit (MU) recruitment and rate coding (rates at which motor neurons discharge action potentials). The authors did not measure this and so it needs to be clear that this is speculative.

The reviewer is correct, it is speculative and a statement made by other researchers. So we made it clearer, and it reads now, “As previously reported in the literature, the force produced by muscles depends on the motor unit (MU) recruitment and rate coding (rates at which motor neurons discharge action potentials)”.

Line 176: “These deficits also might explain the difficulty in sustaining repetitive muscle contraction.” This is not warranted in your document since it was not measured

Same as above. We did not measure it, but we were citing another literature to make this assumption. We made it clearer now as it reads, “These deficits also might explain the difficulty in sustaining repetitive muscle contraction as found in previous research”.

Reviewer 2 Report

Dear Authors

this study is really innovative and well-presented.

The only question is produce an additional table with the difference across groups but also adjusting for confounders such as age, sex, bmi and stage of disease

Author Response

Dear Authors, this study is really innovative and well-presented.

Thanks for your encouraging words.

The only question is produce an additional table with the difference across groups but also adjusting for confounders such as age, sex, bmi and stage of disease

We have created an additional table including the data adjusted by these covariates.

Reviewer 3 Report

  • Only formal names should be capitalized (eg, “postural instability” rather than “Postural Instability.
  • Reduced rate of force production accompany weakness in normal weaker people as well as patients who are weak from pathology (eg, stroke, spinal cord injury, and multiple sclerosis).
  • What was the basis of the sample size?
  • The first column of Table 1 needs a heading. Abbreviations need to be designated.
  • A representative force x time curve from each of the 3 groups would be illustrative.
  • Some research shows power to be no better at force at explaining function (eg, gait).
  • Some key references are missing (eg, Koller and Case, 1986)

Author Response

Only formal names should be capitalized (eg, “postural instability” rather than “Postural Instability.

Thanks for letting us know. We revised the manuscript to reduce these capitalizations.

Reduced rate of force production accompany weakness in normal weaker people as well as patients who are weak from pathology (eg, stroke, spinal cord injury, and multiple sclerosis).

We agree with the reviewer; this statement has been made in the text too.

What was the basis of the sample size?

This study is a secondary analysis from the other two studies; therefore, our sample size was chosen from convenience. We made this clear in the manuscript.

The first column of Table 1 needs a heading. Abbreviations need to be designated.

We included a heading and added appropriate abbreviations. Thanks for noticing these errors.

A representative force x time curve from each of the 3 groups would be illustrative.

We created this figure. Thanks for your suggestion.

Some research shows power to be no better at force at explaining function (eg, gait). Some key references are missing (eg, Koller and Case, 1986)

Thanks for suggesting these references. In the current project, we focused on references specific to PD; therefore, we could better explain the impact of the disease on the force. This is why some key references in power and force production were not used.

Round 2

Reviewer 3 Report

  • Considerable awkward language persists. (An example is the last sentence of the Introduction.) Perhaps a technical writer whose primary language is English should proof-read the work. The authors use the term “derivate.” I believe they mean “derivative.” “There” statements (eg, “there is a debate) should be eschewed. Formal names should be capitalized (eg, UPDRRS) but informal names (eg,peak force) should not be capitalized.
  • Clumn 1 of table 2 needs a label.
  • Abbreviations in figures need designation.
  • How was sample size/power determined.

Author Response

Response: We thank reviewer #3 for the comment, addressing them will certainly improve the quality of the manuscript

Considerable awkward language persists. (An example is the last sentence of the Introduction.) Perhaps a technical writer whose primary language is English should proof-read the work. The authors use the term “derivate.” I believe they mean “derivative.” “There” statements (eg, “there is a debate) should be eschewed. Formal names should be capitalized (eg, UPDRRS) but informal names (eg, peak force) should not be capitalized.

Response: We carefully read the manuscript and we also asked a native English speaking to read and revise the manuscript.

Clumn 1 of table 2 needs a label.

Response: We included the label accordingly.

Abbreviations in figures need designation.

Response: We included the abbreviation in the captions of the figures

How was sample size/power determined.

Response: In the methods section (line 77) we state that “we chose a convenience sample of thirty-eight participants for this study”. As the recruitments of the participants were performed as part of a project which involved different studies, we did not compute a prior sample size calculation specifically for this study. Therefore, we included this information in the limitation section as a recommendation for future studies.

Lines 234-237: “(4) the participants enrolled in this study were recruited from a larger project involving different studies, therefore, the sample size of subgroups was unequal and relatively small (TD n=14 and PIGD n=10 participants).”

Lines 241-242: “(v) future studies should consider larger sample sizes and sample sizes calculation accordingly.”